# 1-Piperidine Propionic Acid Protects from Septic Shock Through Protease Receptor 2 Inhibition

**DOI:** 10.3390/ijms252111662

**Published:** 2024-10-30

**Authors:** Roberto Luisetto, Marco Scarpa, Gianmarco Villano, Andrea Martini, Santina Quarta, Mariagrazia Ruvoletto, Pietro Guerra, Melania Scarpa, Monica Chinellato, Alessandra Biasiolo, Edoardo Campigotto, Daniela Basso, Matteo Fassan, Patrizia Pontisso

**Affiliations:** 1Department of Surgical, Oncological and Gastroenterological Sciences, University of Padova, Via Giustiniani 2, 35128 Padova, Italy; roberto.luisetto@unipd.it (R.L.); marco.scarpa@unipd.it (M.S.); gianmarco.villano@unipd.it (G.V.); 2Department of Medicine, Azienda Ospedaliera-Università Padova, Via Giustiniani 2, 35128 Padova, Italy; andrea.martini@aopd.veneto.it (A.M.); edoardo.campigotto@aopd.veneto.it (E.C.); 3Department of Medicine, University of Padova, Via Giustiniani 2, 35128 Padova, Italy; santina.quarta@unipd.it (S.Q.); mariagrazia.ruvoletto@unipd.it (M.R.); pietro.guerra@studenti.unipd.it (P.G.); monica.chinellato@studenti.unipd.it (M.C.); alessandra.biasiolo@unipd.it (A.B.); daniela.basso@unipd.it (D.B.); matteo.fassan@unipd.it (M.F.); 4Immunology and Molecular Oncology Diagnostics, Veneto Institute of Oncology IOV-IRCCS, Via Gattamelata, 64, 35128 Padova, Italy; melania.scarpa@iov.veneto.it; 5Veneto Institute of Oncology, IOV-IRCCS, Via Gattamelata, 64, 35128 Padova, Italy

**Keywords:** protease-activated receptor 2, septic shock, experimental endotoxemia, 1-piperidine propionic acid

## Abstract

Sepsis is a complex disorder caused by a dysregulated host response to infection, with high levels of morbidity and mortality. Treatment aimed to modulate immune response and maintain vascular function is still one of the major clinical challenges. This study was designed to test the effect of the small molecule 1-Piperidine Propionic Acid (1-PPA) as molecular targeted agent to block protease-activated receptor 2 (PAR2), one of the major modulators of inflammatory response in LPS-induced experimental endotoxemia. In the THP-1 cell line, LPS-induced cytokine expression was inhibited by 1-PPA in a dose-dependent manner. In LPS-injected mice, treatment with 1-PPA was effective in reducing mortality and sepsis-related symptoms and improved cardiac function parameters. After 6 h from LPS injection, a significant decrease in IL-6, IL-1β, and IL-10 was observed in the lung tissue of 1-PPA-treated mice, compared to controls. In these mice, a significant decrease in vasoactive molecules, especially kininogen-1, was also observed, mainly in the liver. Histopathological analysis confirmed typical features of sepsis in different organs and these findings were markedly reduced in mice treated with 1-PPA. These data demonstrate the effectiveness of 1-PPA in protecting the whole organism from sepsis-induced damage.

## 1. Introduction

Sepsis is a complex disorder caused by a dysregulated host response to infection, with high levels of morbidity and mortality [1]. Pathogen-induced hyperinflammation and subsequent immunosuppression, as well as endothelial damage, are the dominant features responsible for the high mortality of sepsis, which cannot be adequately addressed by current therapeutic strategies, such as broad-spectrum antibiotic therapy or fluid resuscitation [2,3]. Hence, there is a strong need to develop a new therapeutic strategy that can modulate the immune response and maintain vascular function. Among the main cellular receptors involved in inflammatory response are the protease-activated receptors (PARs), a subfamily of rhodopsin family G-protein-coupled receptors (GPCRs) that are activated by proteolytic cleavage [4]. PAR1-4 are different receptors which regulate distinct cellular responses involved in coagulation and inflammation [5,6]. Among these, PAR2 has drawn much interest due to its critical role in inflammation. PAR2 is mainly expressed in bronchial, vascular and intestinal smooth muscle cells, leukocytes, and keratinocytes [7]. Activation of PAR2 not only initiates diverse signaling pathways involving mitogen-activated protein kinases (MAPK), extracellular signal-regulated kinase (ERK), c-Jun *N*-terminal kinase (JNK), and p38 kinases, but also activates transcription factors such as nuclear factor-κB (NF-κB) and activator protein-1 (AP-1), resulting in expression and secretion of various inflammatory cytokines, including interleukin-6 (IL-6), interleukin-8 (IL-8), and tumor necrosis factor-α (TNF-α) [8,9,10,11,12,13,14,15]. Multiple serine proteases such as trypsin, mast cell tryptase, and neutrophil proteinase 3 are known to activate PAR2, however, unlike other PARs, PAR2 does not respond directly to thrombin [16], but in a thrombomodulin-dependent manner [17]. In addition, PAR2 has been involved in vascular smooth muscle relaxation. In particular, it has been shown that endothelial PAR2 activation determines murine second-order mesenteric arterioles relaxation as a result of multiple endothelial-dependent mechanisms that are both NO-cGMP-dependent and independent [18].

In line with these findings, different studies in rat models of endotoxin-induced acute renal failure have shown that PAR2 inhibition can normalize LPS-induced elevation of renal endothelin-1 and TNF-α, suggesting that blockade of PAR2 may play a crucial role in treating renal injury, via normalization of inflammation, coagulation, and vasoactive peptides [19,20]. The protective effect of PAR2 inhibition has also been documented in acute liver and lung inflammation models [21,22]. A recent study reported that the compound 1-Piperidine Propionic Acid (1-PPA) is a novel inhibitor of PAR2 that blocks the receptor in an inactive conformation through its binding to an allosteric pocket of the receptor, leading to antagonistic effects of MAPKs signaling [23], widely recognized as a signaling cascade belonging to PAR2 [24,25].

This study aimed to use the small molecule 1-PPA as a molecular targeted agent to block PAR2 in an established mouse model of sepsis induced by LPS.

## 2. Results

### 2.1. Effect of 1-PPA in the THP-1 Cell Line

Preliminary experiments carried out in the monocytic THP-1 cell line showed that in LPS-stimulated cells, PAR2 was markedly induced compared to unstimulated cells, and the addition of 1-PPA was able to revert this effect at very low concentrations (1–10 ng/mL) (Figure 1A). In line with these findings, the LPS-induced cytokine expression, including interleukin-1β (IL-1β), tumor necrosis factor-α (TNF-α), IL-6, tumor growth factor-β (TGF-β), and C-C Motif Chemokine Ligand 2 (CCL-2) was also inhibited in a dose-dependent manner (Figure 1B). The stability of the housekeeper in untreated and LPS-stimulated cells is reported in Appendix A. These results were in agreement with Western blot analysis, where the delayed kinetics of protein expression allowed us to detect in cellular extracts, after 5 h, only the decrease in PAR2 and selected cytokines with the highest dose of 1-PPA (Appendix A).

### 2.2. Sequence Homology Between Human and Mouse Interaction Site of PAR2 with 1-PPA

Sequence and structural homology between human and mouse interaction sites of PAR2 with 1-PPA was taken in consideration in order to assess whether the mouse model could be suitable to test the efficacy of 1-PPA in sepsis control. The protein sequence alignment showed a high homology (Identity 83.5%, Similarity 91.5%, Gaps 1%, Score 1737.0) (Figure 2A). Particular attention was focused on the conservation of the residues previously identified in the interaction with 1-PPA [23] in the human protein. All the residues were conserved also in the mouse model and structurally they were superimposed in the same conformation (Figure 2B).

### 2.3. Experimental Model of Endotoxemia

#### 2.3.1. Preliminary Test for Dose-Finding of 1-PPA

The efficacy of different concentrations of 1-PPA was preliminarily tested in a well-recognized experimental model of murine sepsis induced by intraperitoneal inoculation of LPS [26], monitoring clinical status and survival rate. After 7 hrs from LPS injection, all animals treated with 1-PPA but none of the control mice treated with saline injection were alive (Appendix A). The best 1-PPA concentration was the lowest dose of 190 μMol, where mice after 7 hrs showed mild symptoms, like moderate postural abnormalities and a slightly decreased mobility, ascribed to BCSM (Body Condition Scoring for Mice) 2. The groups of mice treated with the other concentrations of 1-PPA showed a delayed onset of clinical symptoms, reaching at 7 hrs the BCSM 5.

#### 2.3.2. Effect of Early or Delayed 1-PPA Injection

To better mimic the efficacy of treatment in human setting, where the recognition of advanced sepsis leading to septic shock is often a critical clinical point for treatment outcome, the administration of 1-PPA compound was carried out in an “early phase” (at 1 h after LPS injection), corresponding to the onset of symptoms and in a “late phase” (at 3 hrs after LPS injection) where symptoms were overt and survival started to decline. The dose of 190 μMol of 1-PPA was used, since this concentration showed the best protective activity in preliminary results. As shown in Figure 3A, no significant differences were observed in survival curves of LPS-injected mice when 1-PPA was administered 1 h or 3 hrs after the LPS inoculation, since 24-hrs survival for the two groups was 60% and 56%, respectively. On the contrary, none of the untreated LPS-injected mice was alive after 24 hrs, and 100% mortality was achieved in this group within 15 hrs after LPS inoculation.

In line with these results, the symptoms recorded by the BCSM score were similar in mice treated early or late with 1-PPA after LPS injection, especially at 4 and 8 hrs where the score was significantly different, compared to the score of untreated LPS-injected mice of the control group (Figure 3B).

#### 2.3.3. Assessment of Cardiac Function

The cardiac function was assessed in mice treated with 190 μMol 1-PPA or saline solution after 5 hrs from LPS injection, since at this time point different clinical performances were observed (Figure 3B), and the animals were still suitable for the ultrasound procedure. As additional control groups, mice not LPS-injected were treated with 1-PPA alone in the same conditions. The main cardiac functions were assessed by echocardiography.

Stroke volume, cardiac output, fractional shortening, and ejection fraction were significantly reduced in mice treated with LPS (Figure 4A), and treatment with 1-PPA determined a significant improvement of all the parameters of cardiac function, while these were not affected by the injection of 1-PPA alone.

In addition, no significant differences in heart rate were observed among the different groups of mice (Appendix A).

#### 2.3.4. Clinical and Laboratory Testing

LPS-injected mice presented significantly lower body temperature over time compared to the corresponding group of mice treated with 1-PPA, and this profile was in line with the body condition score that was significantly worse in untreated animals (Figure 5A,B). Laboratory testing was taken at 6 hrs from LPS injection, since this was the latest time point where the majority of the animals were still alive, although they presented bad clinical conditions. At this time point, LPS injection determined a marked increase in the percentage of neutrophils and monocytes, associated with a significant decrease in the percentage of lymphocytes, while the administration of 1-PPA determined a significant drop in the percentage of neutrophils and lymphocytes, associated with the highest values of the percentage of monocytes and also of eosinophils, although the absolute figures of these cell types were low (Figure 5C). Prothrombin time was reduced, likely an expression of a hypercoagulable state, in LPS-injected mice, while this parameter almost returned to normal range in mice treated with 1-PPA (Figure 5D). No significant differences were observed in renal function and bilirubin levels in untreated and 1-PPA-treated mice (Figure 5E,F), while treatment with the compound was associated with increased transaminase levels (Figure 5G) as a possible result of increased inflammatory activity, as supported by the increase in CRP (Figure 5H).

#### 2.3.5. Peritoneal Fluid Analysis

The analysis of peritoneal fluid in LPS-injected mice untreated or treated with 1-PPA was assessed at 6 hrs from LPS injection. Direct microscopy was carried out first, then FACS analysis was performed to better characterize the cell phenotype. Direct observation was markedly different, since only LPS-injected and untreated mice presented a reddish and cloudy fluid, rich of cells (Figure 6A,B), while peritoneal fluid of 1-PPA-treated mice was clear. The cell count was significantly higher in untreated mice (Figure 6C).

FACS analysis showed that in LPS-injected mice treated with 1-PPA, only the activated CD8/totalCD8 ratio was significantly lower than in LPS-injected mice (*p* = 0.048) (Figure 6D3), while the activated CD4/totalCD4 ratio showed a trend to be lower in untreated LPS-injected mice (*p* = 0.07) (Figure 6D2). No significant differences of cell distribution between the different groups of mice were observed in the monocytes and neutrophil populations (Figure 6D1). In addition, no significant differences were also observed in CD4/CD8 ratio, nor in total CD4+ and CD8+ cells and in the activated subgroups of CD4+ and CD8+ cells (Appendix A).

#### 2.3.6. Histopathology Evaluation

Histopathological analysis confirmed the development of organ failure due to LPS challenge, and typical features of sepsis [27] were observed in different organs. The liver and the lung showed the worst injury scores, and treatment with 1-PPA determined a significant injury reduction in all organs except the spleen, where a trend toward lower values was nevertheless observed (Figure 7A). As reported in the examples of Figure 7B, LPS injection determined remarkable necrotic tissue damage in the liver. In the kidney, interstitial edema and focal areas of inflammation and hyperemia were observed. In the lungs, a high degree of leukocyte infiltrate and large areas of alveolar and interstitial edema with the presence of spots of hemorrhagic effusions were detected, determining pulmonary hepatization. The heart presented foci of necrosis, interstitial edema, rupture of myocardial fibers, and areas of leukocyte infiltration. In addition, some LPS-injected mice showed mitral and aortic valve damage, characterized by thickening of valvular tissues and aggregates of mixed inflammatory cells. In the spleen, a remarkable expansion of the red pulp with vessel congestion was observed. All these findings were markedly reduced in mice treated with 1-PPA. In particular, cell infiltration, interstitial edema, and vascular congestion were barely detected, leading to histological features close to normal. These results demonstrate that 1-PPA treatment can protect from tissue damage and multi-organ failure induced by LPS in experimental sepsis.

#### 2.3.7. Organ Cytokines and Vasoactive Molecules Expression in LPS-Injected Mice

Since the progression of sepsis is characterized by a profound inflammatory response and hemodynamic decompensation, mainly due to inflammatory-mediated capillary leak and vasodilation [28], we have assessed the profile of inflammatory cytokines and of vasoactive molecules in organs harvested after 6 hrs from LPS injection in untreated mice or in mice treated with 1-PPA.

As shown in Figure 8, the results obtained in the lung at this time point have documented significant increase in IL-6, IL-1β, and IL-10, but not in TNF-α and CCL-2, likely reflecting an earlier increase in these latter cytokines. This could also be the reason for the lack of differences observed in the liver and in the kidney. Interestingly, among vasoactive molecules, a significant increase in inducible nitric oxide synthase (iNOS), bradykinin receptors 1/2 (BR1, BR2), and especially of kininogen-1 (KNG1) was observed in the liver, the latter being significantly increased also in the lung. It is worth to know that kininogen-1 is the precursor of bradykinin, which is a potent vasoactive peptide involved in arterioles vasodilation (via the release of prostacyclin, nitric oxide, and endothelium-derived hyperpolarizing factor) and veins constriction (via prostaglandin F2), leading to increased capillary permeability [29].

These results are in line with the significantly increased expression of PAR2 found in the liver of LPS-injected mice, while 1-PPA treatment determined a significant downregulation of this receptor, reaching levels that were comparable with those observed in control mice (Appendix A). It should be noted that in the kidney, the increase in PAR2 levels detected after LPS injection was not reduced by 1-PPA treatment, but rather further increased, likely as a result of high accumulation of this small molecule beyond its therapeutic efficacy in the excretion phase by renal system. This finding could explain the kidney increase in IL-6 and IL-10, even if not statistically significant and not affecting functional (Figure 5E) and histological (Figure 7A) improvement determined by 1-PPA treatment.

## 3. Discussion

Sepsis treatment is still one of the major clinical challenges, since high levels of morbidity and mortality are typical features of this pathological condition [1]. Pathogen-induced hyperinflammation and endothelial damage are the dominant features responsible for the lethality of sepsis, and current therapeutic approaches are not always sufficient [2,3,30]. Several animal models of endotoxin-induced acute vascular damage have shown that PAR2 inhibition can normalize the LPS-induced elevation of TNF-α, suggesting that blockade of PAR2 may normalize inflammation, coagulation, and vasoactive peptides secretion [20,21]. However, Par1−/−, Par2−/−, Par4−/−, Par2−/−:Par4−/−, and Par1−/−:Par2−/− mice all failed to show improved survival or decreased cytokine responses after endotoxin challenge compared with wild type [31]. Nevertheless, the efficacy of exogenous PAR2 inhibition has also been documented in acute liver and lung inflammation models [21,22]. A recent study showed that 1-PPA is a novel inhibitor of PAR2 that blocks the receptor in an inactive conformation, leading to antagonistic effects of PAR2 signaling cascade [23]. Thus, this study aimed to use the small molecule 1-PPA as a molecular targeted agent to block PAR2 in an established mouse model of sepsis induced by LPS.

The efficacy of different concentrations of 1-PPA was preliminarily tested in mice by intraperitoneal inoculation of LPS [26], while monitoring their clinical status and survival rate. After 7 hrs from LPS injection, all animals treated with 1-PPA but none of the control mice treated with saline injection were alive. The best 1-PPA concentration was the lowest dose of 190 μMol, where mice after 7 hrs showed mild symptoms. The dose was similar to that used by our research group in a previous study [32], and it minimized the liver and renal side effects, while maximizing the effect on sepsis-derived damage. Moreover, to replicate a potential sepsis treatment in a human setting, the administration of 1-PPA was carried out at the onset of symptoms (at 1 h after LPS injection) and when the symptoms were overt and survival started to decline (at 3 hrs after LPS injection). Nevertheless, in these two groups, 56–60% of mice survived at 24 hrs, while at this time point, none of the untreated LPS-injected mice was alive. Similarly, symptoms were milder in mice treated early or late with 1-PPA after LPS injection, while they were more severe in untreated LPS-injected mice. Finally, mice treated with LPS and 1-PPA presented higher body temperature over time compared to the corresponding group of mice treated with LPS alone, and this profile was in line with the body condition score that was worse in untreated animals. All these data indicate that 1-PPA treatment is effective in significantly reducing sepsis-induced mortality and sepsis-related symptoms.

To understand how 1-PPA can halve sepsis mortality and symptoms, we investigated the systemic and organ-specific response to the septic stimulus. To test the effect of 1-PPA on monocytes, the main driver of the inflammatory response, preliminary experiments were carried out in the monocytic THP-1 cell line, which is a human cell line recognized as a reliable model for studying monocytic cells’ response to stimuli such as pathogens, cytokines, and other inflammatory signals [33]. The analysis showed that in LPS-stimulated cells, PAR2 was markedly induced and the addition of 1-PPA was able to revert this effect at a very low concentration. In line with these findings, the LPS-induced cytokine expression was also inhibited in a dose-dependent manner. In cultured monocytic RAW264.7 cells, the EPCR-dependent activation of PAR2 by the ternary TF-VIIa-Xa complex was required for the normal LPS induction of messenger RNAs encoding the TLR3/4 signaling adaptor protein Pellino-1 and the transcription factor interferon regulatory factor 8 [34]. Therefore, we can conclude that the PAR2 pathway is crucial in the sepsis cascade and its blockage can effectively arrest the LPS-driven inflammatory activation.

As recently pointed out by the authoritative study by Takahama et al., in the pathogenesis of sepsis, pairwise effects of tumor necrosis factor plus IL-18, interferon-γ, or IL-1β suffice to mimic the whole impact of sepsis on the different tissues [35]. Therefore, we investigated the cytokines profile and the vasoactive molecules in the different organs in untreated mice or mice treated with 1-PPA after 6 hrs from LPS injection. At this time point, in the lung, we observed significantly lower values of IL-6, IL-1β, and IL-10 in LPS-injected mice treated with 1-PPA compared with untreated LPS-injected mice. Similarly, in 1-PPA-treated mice, we observed significantly lower values of iNOS, bradykinin receptors 1 and 2, and especially of kininogen-1 in the liver. The latter was also observed in the lung, compared with untreated controls. The difference in the cytokines and vasoactive molecule levels among the different organs may simply be due to the presence of a single time point that may miss the whole dynamic situation of LPS-induced sepsis [35]. In any case, the effect of 1-PPA administration in lowering IL-6, IL-1β, and IL-10 may directly impact the cytokines pairs identified by Takahama et al. [35], interrupting the pathogenetic cascade that leads to the final sepsis event.

At a systemic level, in LPS-injected and 1-PPA-treated mice, we observed a significant drop in the percentage of neutrophils and lymphocytes compared to LPS injection alone. The relative increase in eosinophils and monocytes detected in 1-PPA-treated mice is consistent with the restoration of these two populations which in previous studies have been described profoundly reduced in the acute phase of LPS-induced sepsis [36,37]. Moreover, administration of 1-PPA to septic mice reduced the hypercoagulable state, and this observation supports the relevance of our study since this coagulation disorder significantly affects the survival rate of septic patients.

A recent study reported that PAR2 cooperates with platelet-activating factor receptor (PAFR) in mediating neutrophil recruitment, lung inflammation, and macrophage activation [38], supporting our observation that PAR2 inhibition is associated with a lower neutrophil mobilization. Moreover, the definite role of PAR2 in stimulating inflammatory cell recruitment through the cytokines network [39] may explain the drop in circulating lymphocytes after 1-PPA administration. Finally, a previous Japanese study observed that PAR2 blocking peptide suppressed TNF-α elevation and attenuated activation of the coagulation [20]. Therefore, our data and those from the literature confirm that PAR2 blockage can favorably modify inflammatory cell mobilization and coagulation activation in response to LPS-induced sepsis.

In our study, cardiac function parameters were significantly improved by 1-PPA treatment in LPS-injected mice. Moreover, untreated mice presented local necrosis, interstitial edema, rupture of myocardial fibers, and areas of leukocyte infiltration in the heart as well as mitral and aortic valve damage in some. Similarly, in our mice series, histopathological analysis confirmed the development of multi-organ failure due to LPS challenge [29] that was partly reversed by treatment with 1-PPA. Several agents such as S100A8/A9 blockers, carvacrol, and astilbin can protect cardiac function from sepsis-induced damage [40,41]; thus, preserving cardiac function in sepsis is the hallmark of the new antiseptic agent. Therefore, the effect of 1-PPA on the damage to the heart induced by the sepsis appears to be either functional or anatomical and it is probably a key feature in the survival of these mice. It is worth to note that the injection of 1-PAA alone was well tolerated and did not affect cardiac parameters, as 1-PAA-injected mice maintained a well-preserved cardiac function. These results are in line with previous findings obtained in vivo and exploring the toxicity of this compound, where no relevant functional and histological injury was observed in the liver and kidney [32].

In our mice series, the analysis of peritoneal fluid showed that in 1-PPA-treated mice, peritoneal fluid was much clearer than those of LPS-alone treated mice and the cell count was indeed significantly lower in this group of mice, likely as a result of a lower recruitment of immune cells inside the peritoneal cavity after experimental endotoxemia. Moreover, flow cytometric analysis showed that in the peritoneal fluid, in mice treated with 1-PPA, the activated CD8/totalCD8 ratio was significantly lower than in LPS-injected untreated mice. A study by Chan et al. showed that PAR2 activation alters guinea-pig lymphatic vessel contractile and electrical activity via the production of endothelium-derived cyclo-oxygenase metabolites [42] and this phenomenon may explain why the inhibition of PAR2 may lead to a macroscopically different peritoneal fluid. Moreover, a study by Wang et al. observed that TF-PAR2 signaling contributes to the accumulation of hepatic CD8(+) T cells [43], suggesting a possible mechanism of CD8 depletion in 1-PPA-treated mice.

Finally, although Crilly et al. observed that in PAR2 agonist or antagonist-treated mice, spleen histology did not differ between groups [44], in our mice series, we observed a reduction of the expansion of spleen red pulp and vessel congestion in mice treated with 1-PPA, although the difference was not statistically significant. These results suggest that 1-PPA treatment can protect from tissue damage and multi-organ failure induced by LPS in experimental sepsis, providing evidence that PAR2 inhibition is a crucial event in preventing the activation of pathways leading to cytokine release, hypercoagulable state, and vasodilation, which are the hallmarks of septic shock.

The main limits of this study are the lack of human data, that at this stage of the experimentation of the effectiveness of 1-PPA could not be ethically obtained, and the lack of cytokines data at different time points. We acknowledge the need for time-course studies to evaluate the effects of 1-PPA treatment at multiple time points post-LPS injection to better understand the progression of the inflammatory response and to determine the best timing for 1-PPA treatment. Nevertheless, the results that we have reported seem to stand alone in the demonstration of the effectiveness of 1-PPA in protecting the whole organism from sepsis-induced damage. Another important aspect concerns sepsis pathophysiology that may differ between species as a result of different immune system responses. It should be noted, however, that mice and humans share a highly homologous genetic background and that genome-wide transcriptional comparison of humans and mice have found that the resting and activated immune cells of both organisms share a conserved transcriptional program and associated regulatory mechanisms [45]. Anyhow, we are aware that the reproducibility and translational value of available sepsis models are often questioned, due to the highly heterogeneous and complex nature of immune response in human sepsis, which cannot be consistently reproduced in mice.

In conclusion, our study showed that 1-PPA is effective in reducing mortality and symptoms in LPS-induced sepsis. This effect is exerted through the inhibition of a PAR2 driven inflammatory pathway that involves single organ cytokine network, inflammatory cells mobilization and infiltration, together with vasoactive molecules secretion.

## 4. Materials and Methods

### 4.1. Study Design

This study was designed to test the effect of the small molecule 1-PPA in an established mouse model of sepsis induced by LPS. The direct effect of 1-PPA on a macrophage cell line was used to assess the effect of this molecule in reducing the inflammatory response. A mouse model of sepsis with LPS was then used to test the effect of 1-PPA at different time points of sepsis, on cardiac function during the sepsis evolution, on the sepsis-induced organ damage and on the cytokines network. Experimental protocol was reviewed by the Institutional Animal Care and Use Committee (Organismo Preposto al Benessere Animale-OPBA) of the University of Padova and then approved by the Italian Ministry of Health (authorization number 749-2022-PR) according to DLGS 26/2014. This study was performed in accordance with the ARRIVE (Animal Research: Reporting of In Vivo Experiments) Guidelines “https://arriveguidelines.org/ (accessed on 29 October 2024)” about animal laboratory procedures. In particular, mice were randomized and blinding was applied during the allocation phase, in the conduct of procedures. All measures were taken to minimize any pain or discomfort to the animals.

### 4.2. Human and Mouse PAR2 Sequence Alignment

In order to assess whether 1-PPA could be effective also in the mouse experimental model of sepsis, human and mouse PAR2 sequences (respectively Uniprot entry: P55085, P55086) were compared by pairwise alignment performed with EMBOSS NEEDLE server (EMBL) [46] with default parameters.

### 4.3. PAR2 Structural Comparison

The structural transmembrane domain of human and mouse PAR2 was compared using the structure of the human protein (PDB ID: 5NDD) [47], while mouse homologous was obtained from the AlphaFold prediction database (AF-P55086-F1). Proteins were truncated from Val61 to Ser348 and Ile63 to Ser350 for human and mouse structure, respectively. Superimposition of the two proteins and Root Mean Square Deviation (RMSD) were evaluated with UCSF Chimera [48].

### 4.4. Cell Cultures

The monocytic cell line THP-1 was purchased from the American Type Culture Collection (ATCC, Manassas, VA, USA). Cells were cultured in RPMI medium containing 10% fetal bovine serum, 100 U/mL of penicillin, 100 μg/mL of streptomycin (Merck Life Science, Milan, Italy) and 0.05 mM 2-mercaptoehanol (Thermo Fisher, Waltham, MA, USA) at 37 °C in 5% CO_2_ in a humidified atmosphere incubator. To assess the effect of bacterial component on monocytic cell activation in the presence or absence of 1-PPA, THP-1 cells were incubated for 24 hrs with fresh culture medium and were then stimulated with LPS from *Escherichia coli* O111:B4 (10 μg/mL, SIGMA, St. Louis, MO, USA) for 2 or 5 hrs in the presence of different concentrations of 1-PPA (0–1–10–100 ng/mL, SIGMA, St. Louis, MO, USA). Cellular extracts were then used for mRNA isolation and cytokine analysis by molecular amplification and Western blot techniques.

### 4.5. Sepsis Mouse Model

#### 4.5.1. Preliminary Experimental Protocols

C57BL6/J mice, 6–7 weeks of age and weighing 20–25 g, were purchased from Charles River Italia (Calco-Lecco, Italy) and were bred at the Interdepartmental Research Center for Experimental Surgery of the University of Padova (Authorization nr 102/2004-A) in standard condition under 12-hrs light/dark cycles at 22 ± 2 °C and free access to food and water. During the experiments, animals were placed under an infrared lamp to limit hypothermia-related distress. Our study examined male and female animals, and similar findings are reported for both sexes.

In preliminary experiments of dose-finding for 1-PPA, mice were randomly assigned to five groups (n = 6/group) and treated as follows: (1) intraperitoneal (i.p.) injection of 100 μL of saline solution as the negative control group; (2) i.p. injection of LPS from *Escherichia coli* O111:B4 (20 mg/Kg) as the positive control group; (3) i.p. injection of LPS followed after 60 min by 1-PPA i.p. injection (190 μMol in 100 μL of saline solution); (4) i.p. injection of LPS followed after 60 min by 1-PPA i.p. injection (1.9 mMol in 100 μL of saline solution); (5) i.p. injection of LPS followed after 60 min by 1-PPA i.p. injection (19 mMol in 100 μL of saline solution). Concurrently with the stimulus with LPS, to limit important fluid losses following septic shock, each animal was hydrated by subcutaneous injection of 0.5 mL of saline solution. To assess the effects of the experimental treatment, mice were followed up until death and body conditions were recorded every hour according to BCSM (Body Condition Scoring for Mice) as follows: score “0” normal condition; “1” ruffling/coat condition; “2” ruffling/coat condition, decreased physical activity; “3” ruffling/coat condition, decreased physical activity, difficulty breathing; “4” ruffling/coat condition, decreased physical activity, difficulty breathing, abdomen moderately cold to the touch; “5” ruffling/coat condition, decreased physical activity, difficulty breathing, abdomen moderately cold to the touch, lethargy; and “6” was equivalent of death, corresponding to the humanitarian endpoint, where the animal was suppressed [49]. Body condition score assessment was performed blindly by two observers at each scheduled experimental time point.

Since the most effective dose of 1-PPA was that at 190 μMol, subsequent experiments with 1-PPA were carried out at 190 μMol concentration and the investigations are described in detail below.

#### 4.5.2. Evaluation of the Efficacy of 1-PPA Injection at Different Time Points

The efficacy of 1-PPA treatment was assessed by injecting the compound in mice at 1 (n = 6) or 3 hrs (n = 6) after LPS injection in order to simulate the early and late sepsis condition. Mice were observed for 24 hrs to record the body condition score and survival rate according to the previously described criteria.

#### 4.5.3. Assessment of Cardiac Function

The cardiac function in mice (n = 4 untreated controls, n = 4 mice treated with 1-PPA only, n = 8 mice treated with LPS and saline solution, and n = 8 mice treated with LPS followed after 1 h by 190 μMol 1-PPA injection) was assessed using the high-resolution in vivo micro-imaging system for small animals (Vevo 2100) and MS400 transducer (FUJIFILM VisualSonics, SonoSite, Inc., Bothell, WA, USA). Mice were anesthetized using 3% sevoflurane (Abbott, Chicago, IL, USA) with 1 L/min medical oxygen in an anesthesia induction chamber. Once animals were sedated, they were transferred to the echocardiography platform where they were kept under anesthesia using 1.5% sevoflurane with 1 L/min medical oxygen for the entire procedure. Thorax and abdomen of mice were then shaved to allow an adequate ultrasonographic window. The limbs of the mice were then taped down onto the metal ECG leads on the platform which was heated to ensure that the body temperature of the animals was 37 °C throughout the procedure. The temperature was continuously monitored with a rectal thermometer and the heart rate was monitored throughout the procedure from the ECG trace. The following parameters were measured and calculated as indicated elsewhere [50,51,52]:-Stroke volume (SO) = LVEDV − LVESV-Cardiac output (CO) = (LVEDV − LVESV) × HR-Fractional shortening (FS) = [(LVIDd − LVISd)/LVIDd] × 100-Ejection fraction (EF) = [(LVEDV − LVESV)/LVEDV] × 100


*Abbreviations*
*: left ventricular end-diastolic volume (LVEDV), left ventricular end-systolic volume (LVESV), stroke volume (SO), cardiac output (CO), heart rate (HR), fractional shortening (FS), ejection fraction (EF), left ventricular intra diastolic diameter (LVIDd), left ventricular intra systolic diameter (LVISd), volume (V), and diameter (D).*


To eliminate inter-observer variability of Doppler results, all the measurements were recorded by a single operator.

#### 4.5.4. Clinical and Biochemical Analysis

A total group of 14 mice injected with LPS, untreated or treated with 1-PPA 1 h later (n = 7/group), was monitored at 3 and 5 hrs to assess body temperature, using the Vevo 2010 instrument, and the corresponding clinical status. The animals were then sacrificed at 6 hrs to assess the inflammatory profile of peritoneal fluid and blood test analysis. Peritoneal fluid was harvested by washing with 2 mL of PBS and the collected fluid was directly used for FACS analysis or for total and differential white blood cell count to determine the intraperitoneal leukocyte infiltration. Whole blood was collected by cardiac puncture and placed in dedicated test tubes for laboratory testing. Internal organs, including lungs, spleen, heart, liver, and kidneys, were harvested and were in part stored at −70 °C for molecular analysis or formalin fixed and paraffin embedded for histological evaluation. Blood analyses were carried out using automated instruments at the Laboratory Medicine Unit (Sysmex Automated Hematology Analyzer/Transportation units XN series for hematological analysis and Roche Cobas 8000-c702 for biochemical analysis).

#### 4.5.5. Flow Cytometry

Peritoneal exudate from mice was centrifuged and pelleted cells were washed with PBS. The isolated cells were blocked with anti-mouse CD16/CD32 antibody (eBioscience, Waltham, MA, USA) for 15 min and stained with appropriate combinations of fluorochrome-conjugated antibodies for 30 min on ice. Flow cytometric analysis was performed using a FACSCalibur based on CellQuest software version 5.1 (BD-Becton Dickinson, Franklin Lakes, NJ, USA). The antibodies used are summarized in Appendix A.

#### 4.5.6. Western Blot

Cellular extracts from THP-1 cells treated with LPS for 5 hrs in the presence or absence of different concentrations of 1-PPA were loaded on linear gradient polyacrylamide gel (Invitrogen™ Bolt™ Bis-Tris Plus Mini Protein Gels, 4–12%) and run in Bolt™ MES SDS Running Buffer for 30 min at 200 V. Protein bands were then transferred to nitrocellulose membrane in Power Blotter 1-Step™ Transfer Buffer (Thermo Fisher Scientific, Waltham, MA, USA), applying 25 V for 8 min at room temperature. Nitrocellulose membranes were blocked for 1 h in Phosphate-buffered saline solution (PBS) with Tween-20 0.1% (TPBS) and 5% (*w*/*v*) Bovine Serum Albumin (Merck Sigma-Aldrich, St. Louis, MO, USA). Primary antibody incubations were performed at 4 °C overnight. Membranes were washed three times with TPBS and incubated with a secondary antibody, HRP conjugated, for 1 h at RT. PAR2 expression was detected with rabbit monoclonal anti-PAR2 antibody (Abcam, Cambridge, UK) diluted 1:1000, TNF-α and IL-6 with rabbit polyclonal antibody diluted 1:1000 (GeneTex, Alton Pkwy, Irvine, CA, USA), while mouse monoclonal anti-α Tubulin diluted 1:1000 (Merck Sigma-Aldrich, St. Louis, MO, USA) was used as loading control in both analyses. Anti-rabbit (Merck Sigma-Aldrich, St. Louis, MO, USA) and -mouse (KPL, SeraCare, Milford, MA, USA) HRP conjugated antibodies were used as secondary antibodies. Bands were revealed and quantified with chemiluminescent substrate LiteAblot Plus (Euroclone, Pero MI, Italy) using Alliance Q9 Atom (Uvitec, Cambridge, UK) and normalized to loading control.

#### 4.5.7. Molecular Amplification Techniques

Total RNA was extracted from THP1 cells, using Trizol Reagent (Invitrogen, Carlsbad, CA, USA) according to the manufacturer’s instructions. After determination of the purity and integrity of the total RNA, complementary DNA synthesis was carried out from 1 ug of RNA using LunaScript RT SuperMix (New England BioLabs, Ipswich, MA, USA). Quantitative real-time PCR reactions (RT-PCR) were performed according to Luna Universal qPCR master Mix (New England Biolabs, Ipswich, MA, USA) protocol, using the CFX96 Real-Time instrument (Bio-Rad Laboratories Inc, Hercules, CA, USA). The relative expression was generated for each sample by calculating 2^−ΔΔCt^ [53]. Primer sequences used in the study are reported in Appendix A.

Expression of inflammatory and vasoactive genes (IL-6, IL-1β, TNF-α, IL-10, chemokine (C-C motif) ligand 2 or CCL-2, inducible nitric oxide synthase or iNOS, endothelial nitric oxide synthase or eNOS, bradykinin receptor 1 or BR1, bradykinin receptor 2 or BR2, and Kininogen 1) was assessed in mouse tissues, including lung, liver, and kidney, by real-time PCR. Total RNA was extracted from frozen tissue using Total RNA purification plus kit (Norgen Biotek Corp., Thorold, ON, Canada) according to the manufacturer’s instructions. One μg of extracted RNA for each tissue sample was reverse transcribed into cDNA first-strand by LunaScript, RT SuperMix kit (New England Biolabs Inc., USA). The SYBR Green real-time PCR was performed using a Luna Universal qPCR Master Mix (New England Biolabs Inc., USA) and the CFX96 Real-Time instrument (Bio-Rad Laboratories Inc., Hercules, CA, USA).

PCR was performed using specific primers of the mouse designed from the published nucleotide sequences on GenBank^TM^ (Appendix A).

The amplification of specific transcripts was confirmed by melting curve profiles at the end of each PCR cycle using the specific routine built-up in the LightCycler instrument. The relative quantification of the transcripts of each gene of interest was normalized to the housekeeping gene (β-actin). The comparative cycle threshold method (2^−ΔΔCt^), which compares between groups differences in cycle threshold values, was used to obtain the relative fold change of gene expression [53].

#### 4.5.8. Histological Examination

For the histological analysis, the lungs, spleen, heart, liver, and kidneys were harvested. The tissue specimens were fixed in a 4% buffered formalin solution, dehydrated through an ethanol series, embedded in paraffin, and sliced into 5-μm-thick sections. After deparaffinization, the sections were stained with hematoxylin and eosin (HE) using the standard histological method. Slides were examined blindly by two independent investigators using a Leica DM4000B microscope (Leica, Wetzlar, Germany) equipped with a Leica DFC420 camera. Three slides per specimen were used to assess organ injury. The injury scores (from 0 to 4) were assigned on the basis of the characteristics of each organ, following the indications of the Atlas of non-neoplastic lesions for rodents (US Dept. of Health and Human Services, National Toxicology program, Guide for standardizing terminology in toxicologic pathology for rodents “https://ntp.niehs.nih.gov/atlas/nnl (accessed on 29 October 2024)”. In detail, the following parameters were considered: the degree of neutrophil infiltration, alveolar and interstitial edema, and hemorrhage for the lung [54]; degeneration, necrosis, leukocyte infiltrate, and blood stagnation, grade of inflammation, necrosis, edema, hyperemia for the kidney [55]; focal leukocyte infiltrate, angiectasia and necrosis for the liver [56]; number of erythrocytes in the red pulp along with edema, cellular debris and proteinaceous fluid containing fibrin for the spleen [57]; inflammation, necrosis, edema and hyperemia for the hearth [58]. The mean score for each parameter was then used for statistical analysis. The total score of different organs was expressed as the result of the mean of injury scores calculated for each organ.

### 4.6. Statistical Analysis

Data were expressed as mean ± SEM or Median and Interquartile Range (IQR). Statistical analysis was performed using the Mann–Whitney test or Kruskal–Wallis test for continuous variables and the chi-square or Fisher exact test for categorical variables as needed. The survival rates were calculated using the Kaplan-Meier methods and groups were compared using the Log-Rank Test (Mantel Cox). All tests were two-tailed and a *p* value < 0.05 was considered significant. Data in bar graphs represent means ± SEM or Median and Interquartile Range (IQR) and were obtained from at least three independent experiments. Results of the Body Condition Score over time in LPS-injected mice and treated with early or late 1-PPA injection are expressed as median score values (±SEM). Dunn’s multiple comparison test was used to compare the three different groups at each time point.

Data were analyzed by GraphPad Prism 9 (GraphPad Software Inc., San Diego, CA, USA) and SPSS version 29.0 software (SPSS Inc., Chicago, IL, USA).

## 5. Conclusions

The results of our study indicate that 1-Piperidin Propionic Acid reduces mortality and sepsis-related features in septic shock through the inhibition of protease-activated receptor 2, preventing the activation of pathways leading to cytokine release, hypercoagulable state, and vasodilation, which are the hallmarks of septic shock.

## 6. Patents

Italian Patent Application N. 102022000014593 filed by the University of Padova; PTC/IB2023/057138 filed on 12 July 2023.

## Figures and Tables

**Figure 1 ijms-25-11662-f001:**
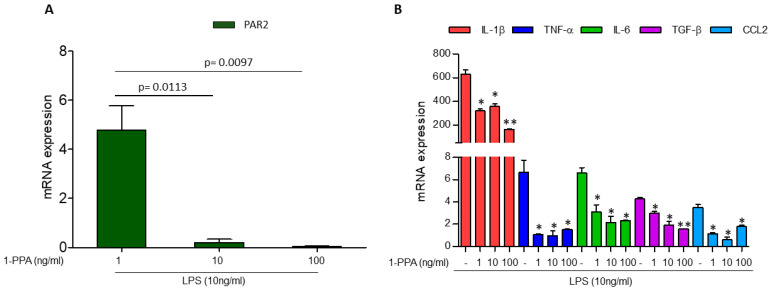
Effect of 1-Piperidin Propionic Acid in LPS-induced PAR2 and cytokine expression in THP-1 cell line. Quantitative real-time PCR of PAR2 (**A**) and of inflammatory cytokine genes (IL-1β, TNF-α, IL-6, TGF-β, and CCL2) (**B**) in the monocytic THP-1 cell line treated for 2 h with LPS (10 ng/mL) in absence or in presence of different concentrations of 1-Piperidin Propionic Acid (1-PPA). Results are expressed as fold increase. * *p* < 0.05, ** *p* < 0.005 compared to untreated samples (Mann–Whitney test).

**Figure 2 ijms-25-11662-f002:**
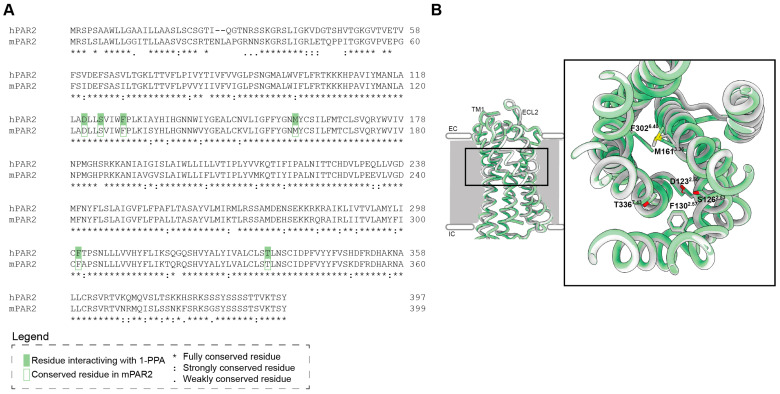
Comparison of human and mouse PAR2 shows conservation in the 1-PPA interaction site. (**A**) Pairwise alignment of human Protease-Activated Receptor 2 and mouse homologous. Green boxes indicate the main residues involved in the interaction between human PAR2 and 1-PPA. Lined boxes refer to residues that are fully conserved in the mouse homologous in the same position. (**B**) Superimposition of PAR2 transmembrane domain; human protein (PDB 5NDD) is represented in white, while murine AlphaFold model (AF-P55086-F1-v4) is represented in green. Zoom shows the residues of mouse PAR2 conserved in the binding of 1-PPA.

**Figure 3 ijms-25-11662-f003:**
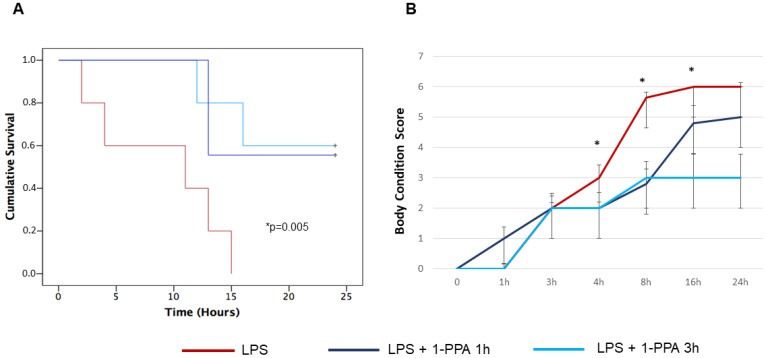
Effect of early and late administration of 1-PPA on LPS-injected mice. (**A**) Kaplan Mayer curves of cumulative survival of mice injected with LPS alone (n = 6), with LPS and treated with 1-PPA 190 μMol after 1 h (early administration, n = 6) or after 3 hrs (late administration, n = 6). * *p* < 0.005 LPS vs. LPS + 1-PPA at 1 or 3 hrs (Log-Rank Mantel Cox test). (**B**) Clinical conditions monitored over time by the Body Condition Score in LPS-injected mice treated with early or late 1-PPA injection. Results are expressed as median score values (±SEM) in different groups. Dunn’s multiple comparison test was used to compare the results in the different groups at each time point (* 4 hrs: LPS vs. LPS + 1-PPA 1 h and LPS vs. LPS + 1-PPA 3 hrs, *p* = 0.005; * 8 h: LPS vs. LPS + 1-PPA 1 h and LPS vs. LPS + 1-PPA 3 hrs, *p* = 0.0004; * 16 hrs: LPS vs. LPS + 1-PPA 1 h: *p* = ns and LPS vs. LPS + 1-PPA 3 hrs, *p* = 0.035).

**Figure 4 ijms-25-11662-f004:**
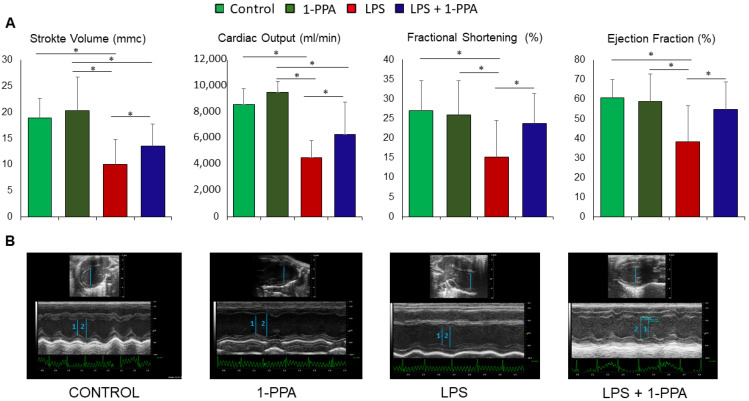
Measurement of cardiac function in LPS-injected mice untreated or treated with 1-PPA. Cardiac function was assessed by ultrasound after 5 hrs in mice injected with LPS and untreated or treated after 1 h with 190 μMol 1-PPA (n = 8/group). Additional control groups of mice (n = 4/group) were not LPS-injected and not treated with 1-PPA (Control) or treated with 1-PPA without previous LPS injection (1-PPA). All measurements were carried out using the Vevo 2100 System (Visualsonics). (**A**) Representation of cardiac parameters in the different groups of mice. Data are expressed as median and interquartile range (IQR). * *p* =< 0.05, Mann–Whitney test. (**B**) Representative examples of cardiac function in mice from the different groups in B-Mode and in M-Mode. CONTROL: untreated control mouse. Blue bars represent ventricular diameter, 1: left ventricular end diastolic diameter, 2: left ventricular end systolic diameter.

**Figure 5 ijms-25-11662-f005:**
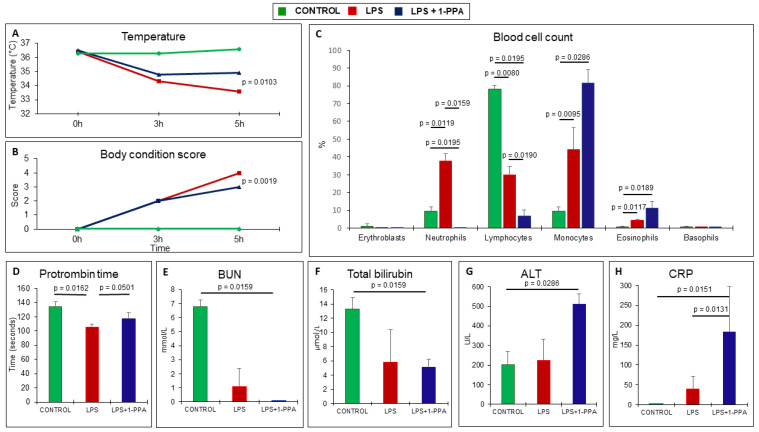
Clinical and laboratory testing over time in LPS-injected mice untreated or treated with 1-PPA. (**A**) Mean rectal temperature profile detected by the VEVO instrument in the different groups of mice. (**B**) Mean of the clinical score monitored over time. (**C**) Percentage of the different types of blood cells detected after 6 hrs in the different groups of mice. (**D**–**H**) Biochemical parameters, including Prothrombin Time (**D**), Blood Urea Nitrogen, BUN (**E**), Total Bilirubin (**F**), Alanine Aminotransferase, ALT (**G**), and C Reactive Protein, C Reactive Protein, CRP (**H**) in the different groups of mice. The results are expressed as Mean values ± SEM (Mann–Whitney test).

**Figure 6 ijms-25-11662-f006:**
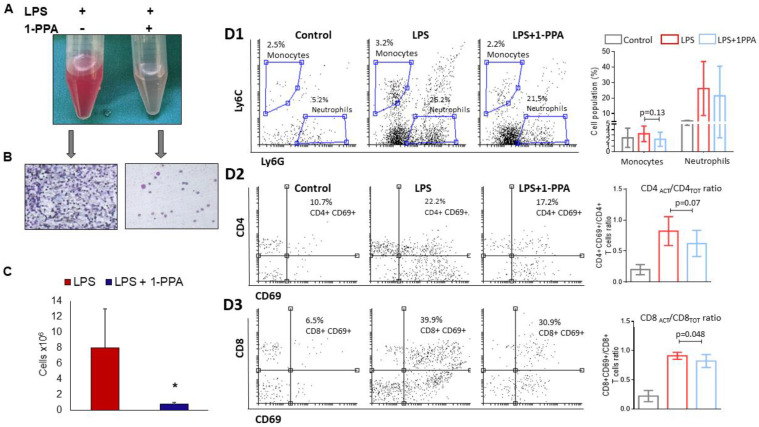
Peritoneal fluid analysis after 6 hrs from LPS injection. (**A**) Example of macroscopic appearance of the peritoneal fluid in an LPS-injected mouse and in an LPS-injected mouse treated with 1-PPA; (**B**) Cytology from peritoneal fluid of the corresponding mice, note the absence of a significant leukocytic component in the sample treated with 1-PPA; (**C**) Cell count from the peritoneal fluid in mice treated with LPS (n = 7) and with LPS + 1-PPA (n = 7). The results are expressed as Mean ± SEM * *p* = 0.0379 (Mann–Whitney test); (**D1**) Neutrophil and monocytes count in the peritoneal fluid. Left panel: cytofluyorimetric analysis (Gate on CD45+ CD11b+ cells); right panel: graphical representation of the Mean values ± SEM (Mann–Whitney test); (**D2**,**D3**) T-lymphocyte subpopulations count in the peritoneal fluid (Gate on CD3+ cells). (**D2**) CD4 T-lymphocytes: Left panel: cytofluyorimetric analysis; right panel: graphical representation of the Mean values ± SEM in activated (ACT) vs. total (TOT) CD4 positive T-lymphocytes (Mann–Whitney test); (**D3**) CD8 T-lymphocytes: Left panel: cytofluyorimetric analysis; right panel: graphical representation of the Mean values ± SEM in activated (ACT) vs. total (TOT) CD8 positive T-lymphocytes (Mann–Whitney test).

**Figure 7 ijms-25-11662-f007:**
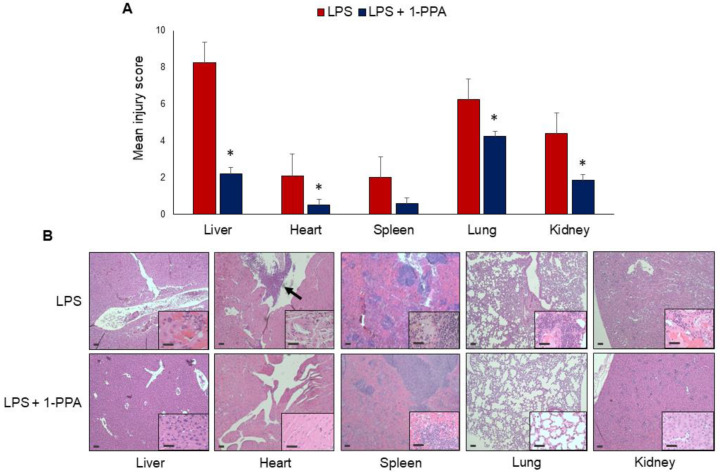
Histological analysis of the organ injury. (**A**) injury score of different solid organs after 6 h from LPS administration in untreated mice (n = 12) and in the group of mice treated with 1-PPA 190 μMol (n = 12) * *p* =< 0.05. The results are expressed as Mean ± SEM (Mann–Whitney test); (**B**) Representative examples of histological results in the different organs, after hematoxylin-eosin staining in one mouse injected with LPS and in one mouse LPS-injected and treated with 1-PPA. The arrow indicates a large area of leukocyte infiltrate in the aortic valve. Magnification of the panels 5× and inserts 20×. Bar = 10 μm.

**Figure 8 ijms-25-11662-f008:**
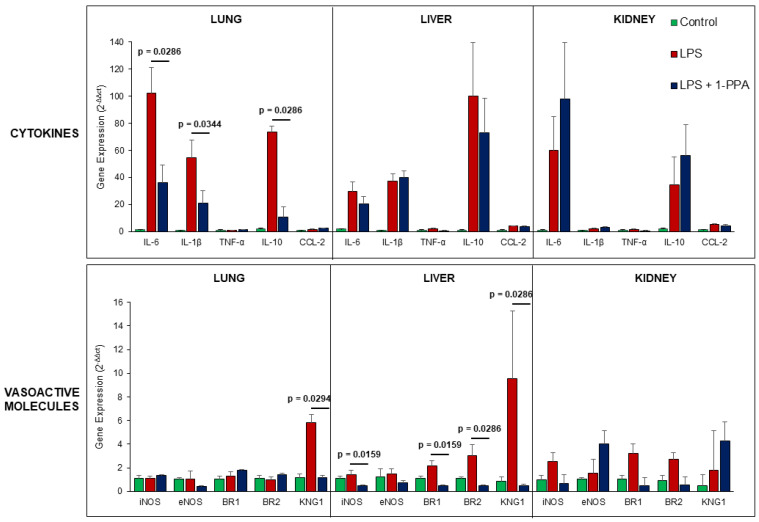
Gene expression of cytokines and vasoactive molecules in different organs in LPS-injected mice. Solid organs were harvested after 6 hrs of LPS administration and saline injection in untreated control mice (n = 6) and in the group of mice treated with 190 μMol 1-PPA (n = 6). Results are expressed as Mean ± SEM of gene expression, reported as 2^−ΔΔct^ relative to basal values. *p* values were reported only for significant differences, *p* < 0.05 (Mann–Whitney test). iNOS: inducible nitric oxide synthase; eNOS: endothelial nitric oxide synthase; BR1: bradykinin receptor1; BR2: bradykinin receptor2; KNG1: kininogen-1.

## Data Availability

The data presented in this study are available upon request from the corresponding author.

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
