# Peer review of "1-Piperidine Propionic Acid Protects from Septic Shock Through Protease Receptor 2 Inhibition"

_ijms, 2024, doi:10.3390/ijms252111662_

Round 1
Reviewer 1 Report
Comments and Suggestions for Authors
This manuscript presents relevant data of potential interest to the scientific community. However, the positive impression is clouded by 1) methodological limitations/imprecisions and 2) an inadequate (graphical) presentation of the data. The text, illustrations, and figure legends also contain a number of inaccuracies. If these limitations are addressed, the data obtained will certainly provide added value to the readership.
Issues related to methodological limitations/imprecisions:
a) Figures 1 and 8: Why was only the mRNA expression analyzed and not the protein expression? This limits the validity of the data. In addition, there are contradictions in the specification of the housekeeper used (beta-actin according to the Methods section, GAPDH according to the Supplement). An additional figure (in the Supplements) is absolutely necessary to prove that the housekeeper used (whichever one) was stable in its expression. Especially under septic conditions after LPS stimulation, this is often not the case, so that a reference to a housekeeper whose expression has been altered by the experimental conditions will lead to the receipt of house numbers instead of real measured values.
b) The time lines used are not explained. Why do the authors think that administration of 1-PPA after 1h is representative of early treatment and administration of 1-PPA after 3h is representative of late treatment? Why are the measurements taken at different times (sometimes after 5h, sometimes after 6h)? Why is there no data for the period >6h? In its current form, it appears that the authors have always chosen the point in time when the most impressive differences could be observed. However, it is important for the scientific community to see the big picture.
c) The individual experiments vary in the number of animals studied (ranging from 4 to 12), without any scientific justification or discussion of a potential resulting limitation.
d) Figure 3: Were measures taken to prevent bias in the assessment of the condition of the mice (i.e. in the assessment of the body condition score)? Blinding or the 4-eyes principle would be useful. The same applies to the determination of the "injury score" in Figure 7.
Issues related to inadequate (graphical) presentation of the data:
a) There is a great deal of variance in the way the data is presented. There are box plots and bar graphs in different versions and visual designs with and without individual values. Standardization is required by selecting the graph type with the maximum information content (see Figure 4).
b) The illustrations are consistently much too small.
c) Figure 2B: There is no indication that the structural model of the murine receptor is merely a prediction that has not yet been validated experimentally.
d) Figure 3B: Revised plot and associated description to include the full 24 hour period. Stopping at time point 7h is misleading as it suggests that the highest body condition score measured in the group of mice treated with 1-PPA is 2. Figure 3D shows that this is not true.
e) Figure 5: What does "basal" mean (corresponds to CNTRL in Figure 4)? What does "clinical score" mean (corresponds to "body condition score" in Figure 3)?
f) Figure 3C+3D: Information on the n-numbers is missing. Figure 6: Information on the n-numbers, the time of sampling and the type of significance test used is missing. In the legend, "cellularity" should probably mean "cell count".
g) Figure 8: The untreated control group is missing.
h) Figure 2 from the Supplements belongs in the main section with its own subchapter.
i) Figures 6 and 8: Insufficient data description in the text. The results of sub-figures 6 D1, 6D2 and 6D3 are not discussed at all. The text for Figure 8 is primarily a cherry-picking of significant effects without describing the overall picture, and also contains a speculative part.
Inaccuracies within the manuscript:
a) Special characters are not displayed correctly throughout. The n-numbers are always not given in the correct form (missing "=").
b) Some of the instruction text on the template was not completely removed (lines 72-73, line 569).
c) Line 87: Really p<0.005 (instead of p<0.001)?
d) Figure 3A: The description of body condition score 3 and body condition score 5 is identical.
e) Line 173: "leukocytes" probably means neutrophils, doesn't it?
f) Line 177: Figure 5D shows the data of prothrombin, not of coagulation time.
g) Lines 178-180: Content correction required. Description does not correctly reflect data in the diagram.
h) Line 526: Missing separation of heading and text.
i) Abbreviations must always be explained in the legend; this is missing in Figure 8, for example.
Comments on the Quality of English LanguageThe manuscript is of good linguistic quality throughout.
Reviewer 2 Report
Comments and Suggestions for Authors
Figure 2: The resolution appears too low, making the labeled text on the protein structure difficult to read. Please provide a higher-resolution version to enhance clarity.
Figure 4B: The M-mode images are difficult to interpret. I recommend cropping the relevant time-lapse sections, enlarging them, and presenting only those in the figure. The full M-mode images can be moved to supplementary materials.
Figure 5: Please explain why an increase in monocytes and eosinophils is observed after the administration of 1-PPA. Even though the absolute cell count remains low, does this shift have a biological impact?
Figure 6D: The titles on the bar graphs are hard to read. I suggest rearranging the figure for better clarity and legibility.
Figure 7: Scale bars are missing from this figure. Please ensure they are included.
Symbols: There seems to be an issue with special symbols (e.g., α, β, Δ). Please check the formatting throughout the manuscript to ensure these are correctly displayed.
Materials and Methods: Please specify the strain of LPS used in the experiment to ensure reproducibility.
Discussion: The discussion on the 1-PPA inhibitor’s effects needs further elaboration. Based on your observations, how does the inhibitor influence the septic mouse model?
Reviewer 3 Report
Comments and Suggestions for Authors
Thank you for submitting your manuscript entitled"1-PIPERIDIN PROPIONIC ACID PROTECTS FROM SEPTIC SHOCK" to the International Journal of Molecular Sciences. Your study addressed a significant clinical challenge in the treatment of sepsis, and your findings on the 1-Piperidin Propionic Acid efficacy in reducing sepsis-induced mortality and symptoms are promising. However, I recommend minor revision of the manuscript to address specific concerns and enhance its clarity and impact. Here are the key issues that need to be addressed:
1. Please mention ethical considerations and welfare measures taken during animal study to align with ethical publishing standards.
2. As you try to simulate early and late sepsis condition in humans, and replicate a potential sepsis treatment in a human settings, beside examination of sequence homology between human and mouse interaction site of PAR2 with 1-PPA , please consider adding discussion about specific aspects of sepsis pathophysiology that may differ between species, such as differences in immune system responses, and how these might impact the generalizability of the findings to human patients.
3. While the paper discusses the specific inhibition of PAR2 by, it does not address the possibility of 1-PPA off-target effects. The potential for off-target interactions is a significant concern that can impact the interpretation of results, so please discuss this issue.
4. Please explain the choice of the THP-1 cell line and its relevance to in vivo findings.
5. The analysis of cytokines and vasoactive molecules at a single time point is important limitation since the progression and resolution of sepsis involve time-based changes in inflammatory and anti-inflammatory processes. Thus, it will be valuable to acknowledge the need for time-course studies to evaluate the effects of 1-PPA treatment at multiple time points post-LPS injection, to better understand the progression of the inflammatory response and determining the best timing for 1-PPA treatment.
I look forward to seeing the final version of your manuscript and believe it will make a significant contribution to the field.
Round 2
Reviewer 1 Report
Comments and Suggestions for Authors
The present revision is a clear improvement of the manuscript. However, the authors failed to optimize the figures. Figure 4 is much too small; in Figure 5, at least the labeling needs to be enlarged. In addition, a standardization of the graphical presentation between the figures is needed. The current mix of variable color, font, and design elements is not acceptable.
Figure 4 also contains a conflicting time indication in the chapter text and legend.
In addition to the poor presentation, the validity of the data is questionable in two respects:
Figures 3B and 5A show a clear difference between the "LPS" and "LPS+1-PPA (3 hours)" groups at time 3h, when the different treatment only begins. Why do mice in the 1-PPA (3 hours) group show a more favorable clinical course almost before the start of treatment?
In Figure S1, the housekeeper GAPDH shows considerable fluctuations in expression. The finding of reduced PAR2 expression is primarily based on this housekeeper and is therefore on very shaky ground. If the Western blot provides insufficient data, why wasn't at least a cytokine determination in the supernatant of the THP1 cells performed by ELISA?
Comments on the Quality of English LanguageEnglish language is fine with only minor corrections needed.
Round 3
Reviewer 1 Report
Comments and Suggestions for Authors
Fortunately, the question marks regarding content and methodology were removed. Nevertheless, the relevant data in the manuscript suffer from inadequate (graphical) presentation. Although the labelling of the graphs has been increased, there is certainly room for improvement in order to allow the reader to engage with the data presented in a pleasant way. The general graphic design of the various figures is also still inconsistent. However, as this is not a matter of technical, content-related improvements, I believe that the instructions for optimisation up to publication are better left in the hands of the journal editors than in the hands of a reviewer.
Comments on the Quality of English LanguageEnglish language is fine with only minor corrections needed.